

# Synergistic approach of hydrometeor retrievals: considerations on radiative transfer and model uncertainties in a simulated framework

Ethel Villeneuve[1], Philippe Chambon[1], and Nadia Fourrié[1]

[1]CNRM, Université de Toulouse, Météo-France, CNRS, Toulouse, France

**Correspondence:** Ethel Villeneuve (ethel.villeneuve@umr-cnrm.fr)

**Abstract.** In cloudy situations, infrared and microwave observations are complementary, with infrared observations being sensitive to the small cloud droplets and ice particles and microwave observations sensitive to precipitation. This complementarity can lead to fruitful synergies in precipitation science (e.g. Kidd and Levizzani, 2022). However, several sources of errors do exist in the treatment of infrared and microwave data that could prevent such synergy. This paper studies several of these sources to estimate their impact on retrievals. To do so, simulations from the radiative transfer model RTTOV v13 are used to build simulated observations. Indeed, we make use of a fully simulated framework to explain the impacts of the identified errors. A combination of infrared and microwave frequencies is built within a Bayesian inversion framework. Synergy is studied using different experiments: (i) with several sources of errors eliminated; (ii) with only one source of errors considered at a time; (iii) with all sources of errors together. The derived retrievals of frozen hydrometeors for each experiment are examined in a statistical study of fifteen days in summer and fifteen days in winter over the Atlantic ocean. One of the main outcomes of the study is that the combination of infrared and microwave frequencies takes advantage of both spectral range strengths leading to accurate retrievals. Each source of error has more or less impact depending on the type of hydrometeor. Another outcome of the study is that even though the errors may decrease the magnitude of benefits generated by the combination of infrared and microwave frequencies, in all cases explored, their combination remains beneficial.

## 1 Introduction

Satellite observations significantly contribute to the quality of numerical weather prediction (NWP). In particular, data in the infrared (IR) and microwave (MW) spectral ranges are widely used to improve weather forecasts (Geer et al., 2017; Chambon et al., 2022). Both spectral ranges are sensitive to water vapor and temperature. In addition, IR frequencies are sensitive to ice crystals and water droplets of the clouds, whereas MW frequencies are also sensitive to solid and liquid precipitation.

All together, this wide range of frequencies are characterised by a significant information content on all hydrometeor phases along the vertical. Therefore, the synergistic use of these frequencies theoretically permits a better description of clouds and





precipitation in NWP models through the assimilation process.

Assimilating all-sky observations usually leads to improvements of resulting weather forecasts of humidity, temperature and also winds thanks to the tracing effect of four dimensional assimilation which infers information on dynamical fields from information on mass fields and conservative quantities. However, this synergistic use within clouds and precipitation has not been achieved yet operationally in any NWP center. While a number of NWP centres operationally assimilate cloudy and rainy

microwave radiances (Geer et al., 2018), this has not been accomplished yet for infrared data although research is definitely ongoing (e.g. Martinet et al., 2013; Geer et al., 2019; Okamoto et al., 2021; Li et al., 2022).

This paper aims at discussing this synergy by performing sensitivity studies on some error sources that could prevent from obtaining positive effects of IR and MW data combination. On the observation modelling side, an important source of uncer-

tainty lies in radiative transfer properties which are not yet consistent across the spectrum (Baran et al., 2014; Eriksson et al., 2018). Indeed, these properties often consider different assumptions for either IR or MW frequencies (e.g. ice crystal shapes, particle size distributions, cloud overlap assumptions, numerical methods to compute the scattering effects). In this study, we will quantify the importance of several of these inconsistencies and compare them to other uncertainties that exist regarding the cloud representation within NWP models. Indeed, microphysical parametrizations and convection schemes make a number

of assumptions which can for instance influence the balance between cloud ice and precipitating frozen particles (e.g. auto-conversion rate from ice to snow) or the balance between cloud liquid water and rain (e.g. auto-conversion rate from droplets to rain). As mentioned above, since IR data are mainly sensitive to cloud ice and MW data to precipitation, an imbalance between the two species in the model compared to observations could lead to spurious effects as well on the synergy. In this study, the impact of these two kinds of inconsistencies on the synergy's ability to retrieve consistent hydrometeor profiles will be studied,

within a one dimensional framework further detailed below.

Satellites from future EUMETSAT missions, the EUMETSAT Polar System (EPS) MetOp Second Generation (EPS-SG) (EUMETSAT, 2013) and Meteosat Third Generation (MTG) (EUMETSAT, 2020) will gather instruments that span IR and MW frequencies: the MTG-Flexible Combined Imager (FCI), an IR instrument, which will provide a high temporal coverage

thanks to its geostationary orbit, the EPS-SG-Ice Cloud Imager (ICI) with submillimetric frequencies ($> 183$ GHz) which will provide observations never acquired before by spaceborne instruments and the MetOp-SG-MicroWave Imager (MWI) with MW frequencies ($< 183$ GHz) inherited from previous instruments. Simulated radiances from these three instruments are considered in this study.

At Météo-France, the current operational method to assimilate MW satellite cloudy and rainy observations in the global NWP model ARPÈGE (Action de Recherche Petite Échelle Grande Échelle) (Courtier et al., 1991; Bouyssel et al., 2022) is called "1D-Bayesian+4D-Var" (Wattrelot et al., 2014; Guerbette et al., 2016; Duruisseau et al., 2017). It consists of a two-step process: (i) a Bayesian inversion that retrieves profiles of hydrometeors and relative humidity, (ii) the assimilation of relative





humidity profiles in the model using a four-dimensional variational (4D-Var) system.


This paper focuses on setting up an experimental framework to use the data of the future instruments mentioned above and to compare the Bayesian retrievals obtained by using either a single-instrument or the three combined.

This study focuses on frozen hydrometeor retrievals since they are associated with larger uncertainties, in terms of radiative

and microphysical properties, than liquid hydrometeors. It aims to quantitatively evaluate the relative importance of some specific radiative transfer (RT) modelling errors across the IR/MW spectrum, with respect to uncertainties within microphysical parameterizations in the NWP model. In section 2, the selected data and methods are presented with details on the inversion algorithm. In section 3, the simulation framework is presented and a number of simulation assumptions are validated. Results are presented in section 4 where errors from either inconsistencies in the RT model or in the NWP model are isolated. Finally,

a discussion is given in section 5.

## 2 Data and Methods

Figure 1 presents the general functioning of the simulated framework defined for conducting the experiments in this study. This framework requires forecasts for both the simulation of observations and the first guess of the inversion ; section 2.1 describes how they are defined. The framework also requires a forward model for the simulation of observations and the inversion

algorithm, described section 2.2. The inversion algorithm is then described in section 2.3 ; the sources of errors introduced both in the forecast model and the forward model are detailed in section 2.4. Finally the validation method for evaluating the quality of the derived retrievals is explained in section 2.5.

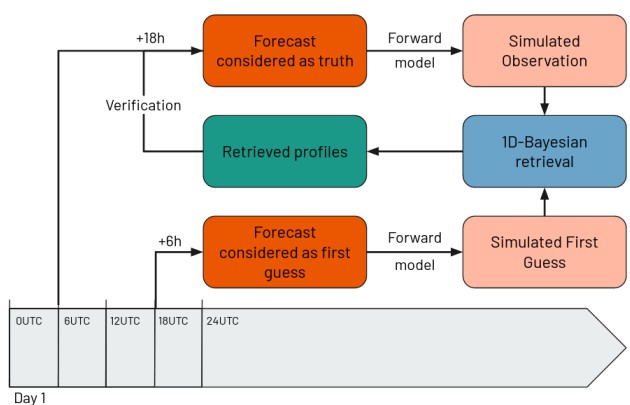

**Figure 1.** Diagram describing the functioning of the methodology employed for the simulated framework



| CLEAR | CLOUDY | PRECIP |
|---|---|---|
| cloud $< 1$ | cloud $> 1$ | cloud $> 1$ |
| precip $< 5$ | precip $< 5$ | precip $> 5$ |

**Table 1.** Hydrometeor's integrated content (g/m$^2$) criteria for each category (clear sky, clouds and precipitation). Cloud content (cloud) stands for the sum of ice and liquid water; Precipitation content (precip) stands for the sum of rain, graupel and snow.

## 2.1 Forecast model

The forecast model used in this paper is the Météo-France's global model ARPÈGE. The spatial horizontal grid is a stretched and tilted grid leading to a variable resolution of 5 km over France and 24 km for the antipodes (South-Western Pacific). The vertical grid is composed of 105 levels from the surface to 0.1 hPa. Regarding the modelling of clouds and precipitation, ARPÈGE makes use of the Lopez (2002) microphysical scheme as well as the Tiedtke/Bechtold convection scheme (Tiedtke, 1989; Bechtold et al., 2008, 2014). Further details on this model configuration, dynamics and physics can be found in Bouyssel et al. (2022).

First guess (FG) and observations (OBS) are both simulated (see subsection 2.2.2) from lagged forecasts valid at the same time (run initialised at 18h UTC with +6h of forecast range for the FG and run initialised at 06h UTC with +18h of forecast range for the OBS). Both forecasts are initialised with the ARPÈGE operational analyses. Using lagged forecasts introduce errors in the localisation and the intensity of clouds and precipitation for the FG with respect to the OBS. In order to validate this framework, a comparison between our simulations and real observations will be performed in section 3 to see if these introduced errors appears to be realistic.

The geographical area of study is located between -60°N and 60°S for latitudes and between -60°E and 60°E for longitudes corresponding to the Meteosat field of view. The full model grid has been thinned by a factor of 4 to prevent the use of too much correlated profiles in terms of error statistics and also to save computing time. FG and OBS are calculated once a day over a 30-day period from 01 to 15 January 2020 and from 01 to 15 June 2020. This period spans both summer and winter seasons to include contrasted meteorological situations without any predominance of seasonal effects in each hemisphere. As a first approach, we have restricted our study to grid points located over sea.

Each profile (OBS and FG) is categorised according to its hydrometeor total column content. Three categories are considered: CLEAR, CLOUDY and PRECIP, built according to their cloud content (ice and water) and their precipitation content (rain and snow) (see Table 1). The selected thresholds correspond to a compromise in order to balance the number of samples in each category. In this study, all cases are taken into account except those where the OBS forecast is CLEAR to avoid retrieving clear-sky values (see table 2, selected cases in italic).





| OBS  FG | CLEAR | CLOUDY | PRECIP |
|---------|-------|--------|--------|
| CLEAR | 251428 (10.88 %) | 114337 (4.95 %) | 22238 (0.96 %) |
| CLOUDY | *107257 (4.64 %)* | *953574 (41.25 %)* | *194793 (8.43 %)* |
| PRECIP | *15612 (0.68 %)* | *169481 (7.33 %)* | *482870 (20.89 %)* |

**Table 2.** Number of points corresponding to each category (clear, cloudy and precipitation) for observations (OBS) and first-guess (FG) over a 30-day period (with the percentage of total cases)

| Wavelength ($\mu$m) | NEdT (K) |
|---------------------|----------|
| 3.8 $\mu$m | 0.2 K |
| 6.3 $\mu$m | 0.3 K |
| 7.3 $\mu$m | 0.3 K |
| 8.7 $\mu$m | 0.1 K |
| 9.7 $\mu$m | 0.3 K |
| 10.5 $\mu$m | 0.1 K |
| 12.3 $\mu$m | 0.2 K |
| 13.3 $\mu$m | 0.2 K |

**Table 3.** Noise Equivalent (NEdT, in K) used as the amplitude of the Gaussian noise applied to simulated observations for each channel and for instrument FCI (EUMETSAT, 2022).

## 2.2 Simulation of future observations

Satellite observations are simulated with the version 13 of the fast radiative transfer (RT) for TIROS Operational Vertical Sounder (RTTOV v13) (Saunders et al., 2020). The settings available in this software allow us to create controlled inconsistencies, by changing parametrizations of crystals shapes and particle size distributions. Compared to RTTOV v12, this version has the specificity to separate the specification of snow and graupel bulk hydrometeors optical properties. In order to generate the observations, the hydrometeor radiative properties used are the latest settings (Geer et al., 2021; Baran et al., 2014; Vidot et al., 2015) that are supposed to best represent the reality. They are implemented for both IR and MW data, as they are assumed to be characterised by the smallest errors with respect to real observations. A Gaussian noise is then added on brightness temperatures (BT) using the Noise Equivalent delta Temperature (NEdT) specifications of each channel of each instrument (see Table 5) to simulate the instrumental noise. In the section below describing the sources of errors introduced, additional details are given for the hydrometeor radiative properties used for the FG.



| Frequency (GHz) | NEdT (K) |
| --- | --- |
| 183.31±7.0 GHz | 0.6 |
| 183.31±3.4 GHz | 0.7 |
| 183.31±2.0 GHz | 0.7 |
| 243.20 GHz | 0.6 |
| 325.15±9.5 GHz | 1.1 |
| 325.15±3.5 GHz | 1.2 |
| 325.15±1.5 GHz | 1.4 |
| 448.00±7.2 GHz | 1.3 |
| 448.00±3.0 GHz | 1.5 |
| 448.00±1.4 GHz | 1.9 |
| 664.00 GHz | 1.5 |

**Table 4.** Noise Equivalent (NEdT, in K) used as the amplitude of the Gaussian noise applied to simulated observations for each channel and for instrument ICI (EUMETSAT, 2013).

## 2.3 Bayesian inversion

In this study, an inversion algorithm is used to perform retrievals of frozen hydrometeors. This inversion method is taken as the same Bayesian algorithm which is used to assimilate microwave cloudy and rainy observations operationally within the ARPÈGE model using retrievals of relative humidity profiles (Guerbette et al., 2016; Duruisseau et al., 2019; Barreyat et al., 2021). Within this framework, each observation is collocated with a First Guess and a surrounding neighborhood (210 km in diameter). From this neighborhood, atmospheric profiles are taken to create an inversion database. A weight is computed for each member of the database. It is calculated from the difference of brightness temperatures $BT$ between OBS and a given FG member, and taking into account observation errors:

$$\text{norm}_\text{neighbor} = \frac{\sum\limits_{\text{channels}} \left( \frac{BT_\text{obs} - BT_\text{neighbor}}{\text{obs\_error}} \right)^2}{\text{nchannels}} \tag{1}$$

$$\text{weight} = \exp(-\frac{1}{2}\text{norm}_\text{neighbor}) \tag{2}$$

for each neighbor profile.

A retrieved profile, hereafter named RET, is then defined as a weighted mean of the inversion database. The corresponding brightness temperature $BT_{RET}$ is also derived from the weighted mean of the inversion database.

This method allows to select channels both in the IR and MW, either separately or together to constrain the inversions. As a first approach, only vertically polarised channels are used for the microwave instruments. Therefore, several channel selections





| Frequency (GHz) | NEdT (K) |
|---|---|
| 18.70 GHz | 0.7 |
| 23.80 GHz | 0.6 |
| 31.40 GHz | 0.8 |
| 50.30 GHz | 0.7 |
| 52.61 GHz | 0.7 |
| 53.24 GHz | 0.7 |
| 53.75 GHz | 0.7 |
| 89.00 GHz | 0.8 |
| 118.75±3.2 GHz | 1.2 |
| 118.75±2.1 GHz | 1.2 |
| 118.75±1.4 GHz | 1.2 |
| 118.75±1.2 GHz | 1.2 |
| 166.90 GHz | 1.1 |
| 183.31±7.0 GHz | 1.0 |
| 183.31±6.1 GHz | 1.1 |
| 183.31±4.9 GHz | 1.1 |
| 183.31±3.4 GHz | 1.1 |
| 183.31±2.0 GHz | 1.2 |

**Table 5.** Noise Equivalent (NEdT, in K) used as the amplitude of the Gaussian noise applied to simulated observations for each channel and for instrument MWI (EUMETSAT, 2013).

have been made: FCI will refer to the selection of each of its infrared channels; ICI will refer to the selection of each of its vertically polarised channels; MWI will also refer to the selection of each of its vertically polarised channels; COMB (combined) will refer to the selection of all vertically polarised channels of ICI and MWI plus the FCI selection.


To determine the observation error that will be used in the Bayesian inversion, *a posteriori* diagnostics (Desroziers et al., 2005) have been used. This diagnostic allows to estimate optimal observation errors. It is computed with the following equation.

$$D = \sqrt{(BT_{OBS} - BT_{FG}) \times (BT_{OBS} - BT_{RET})} \tag{3}$$

with $BT$ the brightness temperature, $OBS$ the observation, $FG$ the first guess used in the inversion framework and $RET$ the
retrieval from Bayesian inversion.

As recommended in Desroziers et al. (2005), several iterations of $D$ calculation have been performed in order to ensure that the *a posteriori* diagnostic converges towards optimal values. The first iteration was set to NEdT value (see Table 5). After



| Wavelength ($\mu$m) | D (K) |
| --- | --- |
| 3.8 | 1.64 |
| 6.3 | 1.09 |
| 7.3 | 1.24 |
| 8.7 | 1.81 |
| 9.7 | 1.28 |
| 10.5 | 2.03 |
| 12.3 | 1.98 |
| 13.3 | 1.57 |

**Table 6.** Desroziers diagnostic $D$ used as the inversion's observation error of each channel and for instrument FCI.

| Frequencies (GHz) | D (K) |
| --- | --- |
| 183.31±7.0 | 1.15 |
| 183.31±3.4 | 1.17 |
| 183.31±2.0 | 1.2 |
| 243.20 | 1.43 |
| 325.15±9.5 | 1.67 |
| 325.15±3.5 | 1.62 |
| 325.15±1.5 | 1.66 |
| 448.00±7.2 | 1.69 |
| 448.00±3.0 | 1.7 |
| 448.00±1.4 | 1.81 |
| 664.00 | 2.29 |

**Table 7.** Desroziers diagnostic $D$ used as the inversion's observation error of each channel and for instrument ICI.

three iterations over the full set of profiles, the derived values only vary marginally ($\mathcal{O}(10^{-2}K)$), therefore the values derived

from this third iteration are taken as the final values which will be used in the rest of the study, they are listed in Table 8.

## 2.4 Source of errors

Several experiments have been conducted in order to document the impacts of possible sources of errors. Two of them are considered in this study: inconsistencies in RT model (and more specifically hydrometeor radiative properties) and errors in the forecast model's microphysical parameterizations. Note that other sources of errors, related to the geometry of observation of

the different instruments, which include their spatial resolution, are not taken into account in this study but could be considered





| Frequencies (GHz) | D (K) |
|---|---|
| 18.70 | 1.09 |
| 23.80 | 1.44 |
| 31.40 | 1.41 |
| 50.30 | 0.85 |
| 52.61 | 0.61 |
| 53.24 | 0.59 |
| 53.75 | 0.58 |
| 89.00 | 1.57 |
| 118.75±3.2 | 1.06 |
| 118.75±2.1 | 1.03 |
| 118.75±1.4 | 1.01 |
| 118.75±1.2 | 0.99 |
| 166.90 | 1.53 |
| 183.31±7.0 | 1.46 |
| 183.31±6.1 | 1.5 |
| 183.31±4.9 | 1.49 |
| 183.31±3.4 | 1.49 |
| 183.31±2.0 | 1.55 |

**Table 8.** Desroziers diagnostic $D$ used as the inversion's observation error of each channel and for instrument MWI.

in a future framework.

The control experiment, hereafter named noERR (no error), refers to the use of an operational forecast without any pertur-
bation as well as without any RT errors. This experiment serves as a baseline, presumably providing the best retrievals, to be
compared to the others to identify which difference leads to the predominant errors in the retrievals. In the following experi-
mental settings, OBS is kept unchanged from the noERR experiment and perturbations are only introduced to the FG used for
the Bayesian inversion and to the RT for the $BT$ simulations with the FG. Information on the perturbations introduced in FG
is given hereafter.

**2.4.1 Introduction of differences in radiative transfer.**

Parameters in RTTOV v13 for the frozen hydrometers are modified in this experiment named mRT (modified RT). Differences
in parameterization and scheme used between noERR and mRT are given in Tables 9 and 10. In the RT for MW, a different
particle shape is used between noERR and mRT, respectively following settings from Geer et al. (2021) and Saunders et al.





(2018), for each hydrometeor. The modified configuration corresponds to the previous default settings of RTTOV-SCATT V12,
defined by Geer and Baordo (2014). In the RT for IR, a different scheme for radiative properties is used for the ice phase,
the one from Vidot et al. (2015) for noERR and the one from Baum et al. (2011) and Wyser and Yang (1998) for mRT, as
suggested in the previous version of RTTOV V12 (Saunders et al., 2018). The use of previous settings for mRT allows a
reasonable representation of hydrometeors but should also bring significant differences from the ones chosen for noERR.

|  | Ice water | Liquid water | Graupel | Snow | Rain |
|---|---|---|---|---|---|
| noERR (Geer et al., 2021) | Large column aggregate (ARTS) | Sphere (Mie) | Column (ARTS) | Large plate aggregate (ARTS) | Sphere (Mie) |
| mRT | Sphere (Mie) | Sphere (Mie) | Sector snowflake (ARTS) | Sector snowflake (ARTS) | Sphere (Mie) |

**Table 9.** Modification introduced in the radiative transfer model settings for microwave instruments in terms of particle shape (Scattering type) from database ARTS (Eriksson et al., 2018).

### 2.4.2 Introduction of differences in the forecast model.

In the forecast model, a number of sub-grid scale processes are parameterized. In particular, those governing the representation
of clouds and precipitation (microphysics of the large-scale precipitation scheme, deep moist convection scheme) both require
the specification of a significant number of tunable parameters. For this study, these parameters are perturbed, based on the
settings used in the ensemble prediction system (EPS) of the ARPÈGE global model of Météo-France (Descamps et al., 2014).
The experiment will be named mMOD (modified model). The use of these specific settings provide a realistic scheme as
they were chosen for their ability to reproduce model errors. With the ARPÈGE EPS, the Random Perturbed Parameter (RPP)
method is used. It consists of perturbing several parameters replacing the default values used in noERR by a random one
selected within a specific range (uniform distribution). The list of perturbed parameters and the default values for the noERR
experiment together with the range of perturbations used in the mMOD experiment are given in Table 12. For generating the

|  | Ice water | Liquid water |
|---|---|---|
| noERR | Baran (Vidot et al., 2015) | Optical Properties of Aerosols and Clouds (OPAC) (Hess et al., 1998) |
| mRT | Space Science and Engineering Center (SSEC) (Baum et al., 2011) (Wyser and Yang (1998) for effective diameter) | Optical Properties of Aerosols and Clouds (OPAC) |

**Table 10.** Modification introduced in the radiative transfer model settings for infrared instrument in terms of particle size distribution schemes.





perturbed model FG, the forecast model was rerun everyday from the operational analysis, with a new set of perturbations. The
same value is used for all grid points for each date.

| parameter - hydrometeor | noERR | mMOD |
|---|---|---|
| sedimentation velocity - cloud ice | 0.08 | [0.01, 0.2] |
| sedimentation velocity - cloud water | 0.02 | [0.005, 0.15] |
| sedimentation velocity - snow | 1.5 | [0.8, 2.2] |
| sedimentation velocity - rain | 5 | [3, 7] |
| auto-conversion - cloud ice - snow coefficient | 0.0035 | [0.0005, 0.006] |
| auto-conversion - cloud water - rain coefficient | 0.001 | [0.0005, 0.006] |
| auto-conversion - minimum ice content (stratiform ice) | 2E-7 | [1E-8, 3E-7] |
| auto-conversion - maximum ice content (stratiform ice) | 3E-5 | [1E-5, 5E-5] |
| auto-conversion - critical water content (stratiform water) | 2E-4 | [5E-5, 1E-3] |
| coefficients - accretion | 1.0 | [0.5, 1.5] |
| coefficients - stratification and ice aggregation | 1.0 | [0.5, 2.0] |
| coefficients - aggregation | 0.2 | [0.1, 1.5] |
| coefficients - calculation of water/ice partitioning | 0.5 | [0.4, 1.0] |
| coefficients - calculation of relative humidity for Smith scheme | 0.5 | [0.5, 0.9] |
| coefficients - calculation of critical relative humidity | 0.3 | [0.3, 1.0] |
| coefficients - calculation of cloud liquid water into rain conversion | 0.004 | [0.002, 0.006] |
| coefficients - maximum evaporation rate for stratiform precipitation | 0.2E-6 | [0, 1E-6] |

**Table 11.** noERR's default value and mMOD range of perturbation (random value between $[X\_MIN, X\_MAX]$) for each perturbed
parameter in microphysics paramterization.

### 2.4.3   Introduction of perturbations in both RT model and forecast model.

A third experiment, named mALL, gathers both differences introduced above. The radiative transfer model used in the inversion
framework is perturbed as in mRT and the microphysical schemes is also perturbed in the forecast model as in mMOD. This

| parameter - hydrometeor | noERR | mMOD |
|---|---|---|
| convection - downdraft mass flux | 0.15 | [0.14, 0.2] |
| convection - entrainment rate | 0.00175 | [0.0016, 0.0019] |
| convection - detrainment rate | 0.000075 | [0.00005, 0.0001] |

**Table 12.** noERR's default value and mMOD range of perturbation (random value between $[X\_MIN, X\_MAX]$) for each perturbed
parameter in the convection parameterization. The perturbation equations are available in Descamps et al. (2014).





experiment will help to understand what kind of inconsistency predominates over the others when both are present, which is

likely to be the case with real observations.

| Experiments | noERR | mRT | mMOD | mALL |
|---|---|---|---|---|
| Characteristics | - Control experiment with the most realistic settings. <br> - Settings for OBS are the same as for FG. | - Differences introduced in RT model for shapes and ice scheme (see Tables 9 and 10). <br> - Microphysics parameterization settings unchanged. | - RT model settings unchanged. <br> - Differences introduced in NWP model (see Table 12). | - Differences introduced in RT model for shapes and ice scheme (see Table **??**). <br> - Differences introduced in NWP model (see Table 12). |

**Table 13.** Characteristics of the First Guess (FG) used to simulate $BT$ in the inversion framework for each experiment. Note that OBS uses the settings of noERR for all experiments.

## 2.5   Metrics

One strength of a fully simulated framework is that the errors of retrievals can be accurately quantified because the truth is known, without the need of specific validation data.

### 2.5.1   Standard deviation.

Errors on retrievals are quantified using standard deviation (std) in the model space. The bias will not be shown as it is overall smaller than the std values in most of the experiments. The std of the inversion error using the simulated observation as reference $(OBS-RET)$ $(STD)$ for each instrument and the combined one allows to know if the combination of all frequencies provides a better retrieval than a retrieval from a single-instrument's inversion.

### 2.5.2   Significance test.

In order to quantify if differences between std from two experiments are significant, a Levene's test (Levene, 1960) is applied with a 95% confidence level (see Appendix A). If the p-value of the two data sets is below 95%, the differences of std are considered as significant.

### 2.5.3   Quantifying error related to perturbations.

The impact of the perturbations introduced in RT and NWP models on the retrievals is quantified with the difference between

the standard deviation of errors the combined retrievals (superscript $c$ for combined) with respect to the standard deviation of errors of the single-instrument retrievals (superscript $i$).





$$DIFF_{mEXP} = STD^c_{mEXP} - STD^i_{mEXP} \tag{4}$$

with $STD$ the standard deviation of the inversion error, mEXP the experiment with perturbations in a model (either mRT, mMOD or mALL).


A negative value means that the retrievals with single-instruments are less accurate than the retrievals with combined-instruments and the more negative the value is, the better the improvement brought by the combined inversion is. A positive value means that the single-instrument provides better results than the combined inversion. The more positive the value is, the larger the degradation due to the combination is.

## 3  Simulated framework validation

(i) As the study is based on simulations both for observations and first guess, a validation metric is needed. To validate the framework, including the lagged forecast assumption, as well as the modifications introduced in the forecast and RT models, an analysis of the FG departure (OBS-FG) distributions has been performed. In order to document the characteristics of these distributions within clouds and precipitation, the std of first guess departure has been computed as a function of a symmetric cloud amount, as originally suggested by Geer and Bauer (2011) for all-sky MW radiance assimilation. The idea is to use a proxy in observation space, which can be computed both for the simulated observations and the first guess. The average of the two, or so called symmetric cloud predictor is then used to categorise the first guess departures.

(ii)For the IR data, we use as cloud predictor the symmetric Cloud Amount ($CA$) defined as (Okamoto et al., 2021):

$$CA = \frac{|BT_{FG} - BT^{clr}_{FG}| + |BT_{OBS} - BT^{clr}_{FG}|}{2} \tag{5}$$

with $BT_{FG}$ FG's brightness temperature, $BT^{clr}_{FG}$ FG's brightness temperature in clear sky, $BT_{OBS}$ OBS's brightness temperature. We will compare $CA$ from FCI onboard MTG (future radiometer) data against the resulting $CA$ of Advanced Himawari Imager on Himawari-8 (HIMAWARI-8/AHI) (current radiometer) that can be found in Okamoto et al. (2021) to validate the hypothesis chosen for simulations.

(iii) For the MW data, we use as cloud predictor the Symmetric Cloud Predictor ($SCP$) (Geer and Bauer, 2010). It is defined 225 as:

$$\begin{cases} P37_{FG} = \dfrac{BT^v_{FG} - BT^h_{FG}}{BT^{v_{clr}}_{FG} - BT^{h_{clr}}_{FG}} \\[2ex] P37_{OBS} = \dfrac{BT^v_{OBS} - BT^h_{OBS}}{BT^{v_{clr}}_{OBS} - BT^{h_{clr}}_{OBS}} \end{cases} \tag{6}$$



with $BT$ the brightness temperature, $FG$ the first guess, $OBS$ the observation at 37 GHz vertically $v$ or horizontally $h$ polarized. $P37$ is the predictor for this window channel.

$$\begin{cases} C37_{FG} = 1 - P37_{FG} \\ C37_{OBS} = 1 - P37_{OBS} \end{cases} \tag{7}$$


$$SCP = \frac{C37_{FG} + C37_{OBS}}{2} \tag{8}$$

Here, we consider the closest channel from 37 GHz available for MWI, which is 31.4 GHz. In this study, this metric is used to compare the perturbations introduced in MWI (future radiometer) simulations against GMI (current radiometer) data in order to verify the chosen settings of simulations.


Figures 2 and 3 shows the results in terms of STD FG departures for one IR channel (10.5 $\mu$m) and one MW channel (89 GHz).

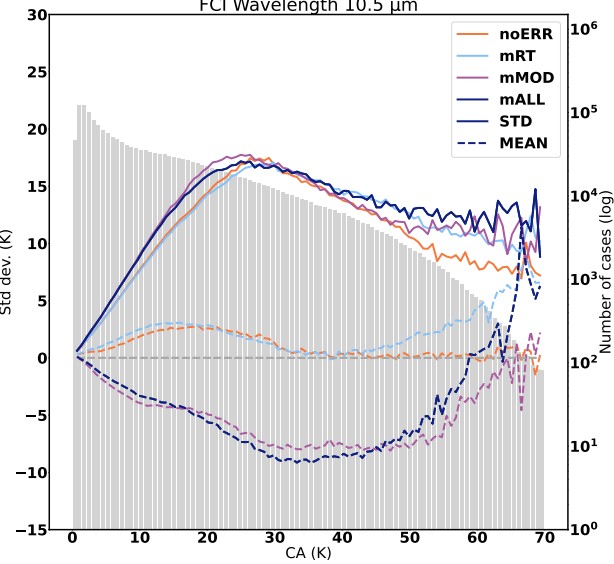

**Figure 2.** Standard deviation (solid line) and average (dashed line) of first guess departures categorised by cloud predictor amount ($CA$) for 10.5 $\mu$m channel of FCI for the different experiments (in color) calculated over the 30-day period, including 15 days in summer and 15 in winter. Histogram represents the number of observations in each category of the cloud parameters.

(iv) For the 10.5 $\mu$m band of the IR instrument FCI (Figure 2), the STD of FG departures increases with the symmetric cloud amount up to 20 K, with only little variations between experiments. On the other hand, the bias from the experiment





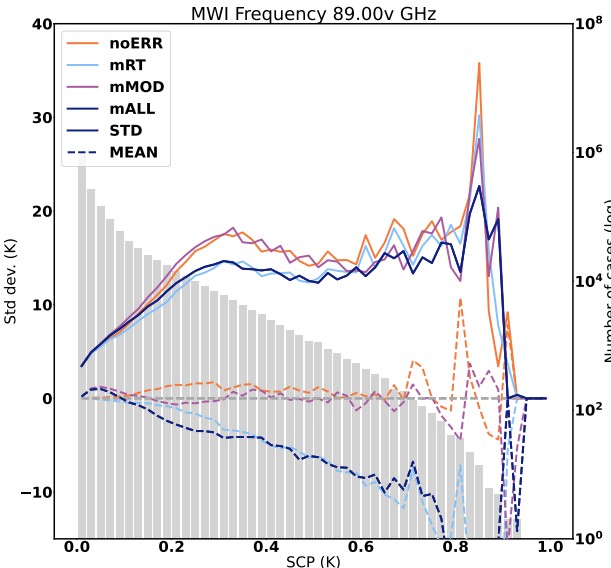

**Figure 3.** Standard deviation (solid line) and average (dashed line) of first guess departures categorised by Symmetric Cloud Parameter ($SCP$) for the 89GHz channel of MWI (b) for the different experiments (in color) calculated over the 30-day period, including 15 days in summer and 15 in winter. Histogram represents the number of observations in each category of the cloud parameters.

with modifications from the model shows significant changes compared to the experiments with no errors or only radiative transfer errors. Comparing those results to the ones of Okamoto et al. (2021) (see their Figure 9 (d) an equivalent band for HIMAWARI-8/AHI channel 13), the simulated framework provides very comparable results in terms of magnitude and error evolution as function of the symmetric cloud predictor.

(v) For the window channel at 89 GHz of MWI (Figure 3), the STD of FG departures also increases with the symmetric cloud amount up to 20 K. Comparing those results to the ones of Lean et al. (2017) (see their Figure 6h for an equivalent channel of the Global Precipitation Measurement (GPM) Microwave Imager (GMI) instrument), the simulated framework provides as well very comparable results in terms of magnitude and error growth as function of the symmetric cloud predictor.

## 4 Results

In this section, the results for each experiment are shown for the following variables: Cloud Ice Water refers to the frozen cloud particles, whereas graupel stands for convective frozen precipitation and snow stands for stratiform frozen precipitation




as defined in Geer et al. (2021). For each species, the effect of combining instruments will be analysed first, then error sources will be added and the effect of the combination reassessed.

## 4.1 Impact of combination and perturbations on CIW.

CIW is an input variable of the RT model for both infrared and microwave spectra. Infrared sensors are expected to perform well for the retrieval of CIW, in particular for thin and non-precipitating clouds as these wavelengths provide accurate cloud top information (Martinet et al., 2014).

### 4.1.1 Impact of infrared and microwave combination

Results of standard deviation of single-instrument and combined-instrument inversion error are provided in Figure 4. This provides information on which instrument retrieves the best CIW with the information content from the simulated observation.

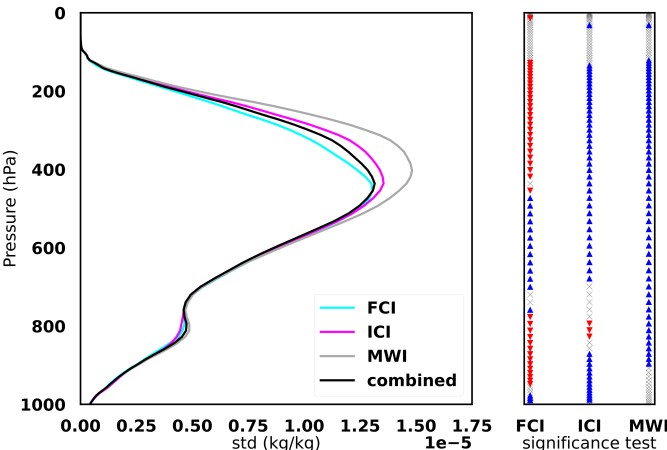

**Figure 4.** On the left, standard deviation of inversion error for CIW as function of pressure (hPa). On the right, Levene's significance test of differences between std of combined and each single instrument. The cyan curve corresponds to the retrieval with FCI only, the magenta curve with ICI only, the grey curve with MWI only, the black curve is the combined retrieval. Blue (resp. Red) arrows indicate a significant improvement (resp. degradation) due to the combination of the three instruments. Grey crosses indicate a non-significant difference.

FCI leads to the best retrieval in the upper layers (200 - 500 hPa) whereas the MW instruments perform better at lower layers (500 - 800hPa) (Figure 4). The combination of all instruments leads to a significant decrease of the std compared to std of 265 single instruments, except for the FCI above 500 hPa which remains the best retrieval. Below 500 hPa, IR is performing less well due to the opacity of clouds and the combined inversion significantly improves all single-instrument inversions down to 800 hPa. The combined inversion gathers the strengths of each spectral domain by taking advantages of IR in the upper layers





of clouds and getting advantages of MW in the lower layers of clouds where the IR-based retrieval weakens in accuracy.

### 4.1.2 Impact of perturbations on synergy

On Figure 5, different information are reported in order to analyse the impacts of perturbations on the instrumental synergy. First of all, the full lines correspond to the differences between the STD of combined retrieval and the STD of single instrument retrieval shown above on Figure 4. This difference is in blue for FCI, red for ICI and green for MWI. On each of the subfigures of figure 5, the full lines are the same. Then on the top of this, the dashed lines depict the same statistics but with the perturbations introduced. The left figure shows the impact of mRT perturbations, the middle figure the impact of mMOD perturbations, and the right figure the impact of mALL perturbations. The colored areas highlight the overlaying of the curves: the baseline colors are cyan for noERR, yellow for mRT, magenta for mMOD and grey for mALL, then color mixing appear with the overlaying, e.g. green when cyan for noERR and yellow for mRT overlay. When the dashed lines are overlayed with the full lines, only the mixed color areas appear and this means that the perturbations have almost no impact on the synergy. Whereas when the dashed lines are overlayed with the full lines, and baseline color areas appear, it means there is an impact of the perturbation on the synergy.

One can see on the left side of the figure 5 that a majority of areas tends to be green for cloud ice which indicates that the modifications of the RT model have rather small impacts on the synergy except for CIW retrieval with ICI. In some cases, a few counter-intuitive results are found, the differences of STD if more negative for the mRT experiment than for the noERR experiment, which means that the synergy is more efficient in the presence of radiative transfer error. This is likely due to an error compensation effect which will need to be further explored.

mMOD leads to a negative contribution to the combined retrievals. Indeed, it can be seen for CIW that the blue area does not overlay with the cyan area: this means that the improvements from the IR-MW combination are reduced. The shape of the mALL curves on the right panel being similar to the mMOD ones, this confirms that the model perturbations therefore lead to more differences in CIW retrievals than the ones in RT model.

### 4.2 Impact of combination and perturbations on snow.

Results for snowfall are shown in Figures 6 and 7.

### 4.2.1 Impact of infrared and microwave combination

The statistics reveal that snowfall is best retrieved by MWI below 700 hPa as expected (Figure 6). The combined inversion provides significantly better results than the three single instrument inversions with the noERR experiments.



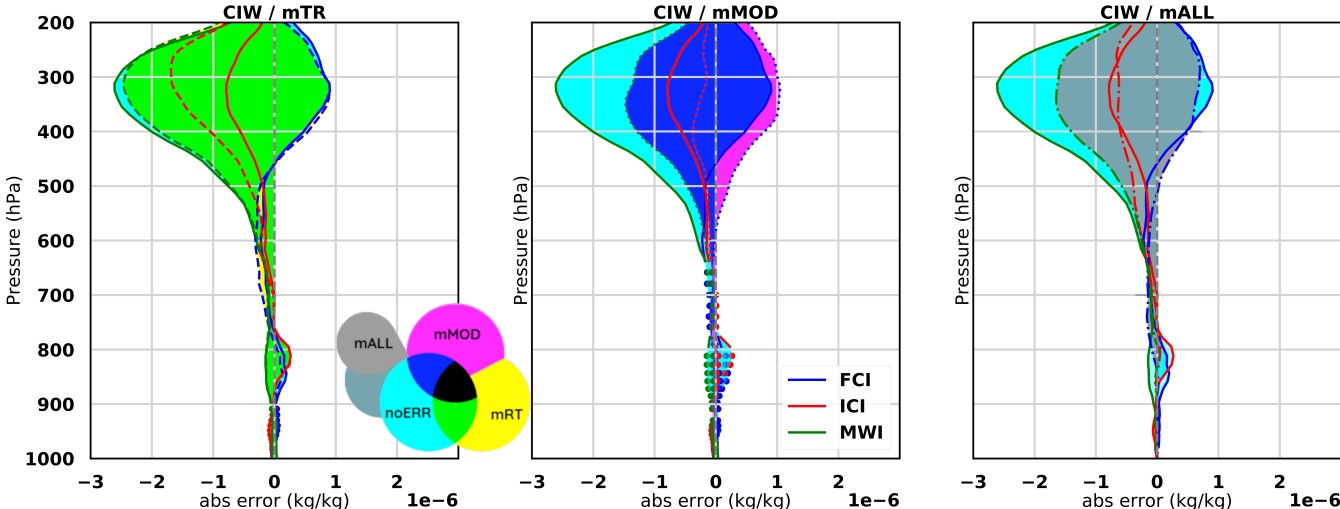

**Figure 5.** Differences of STD between the combined retrievals and the single instrument retrievals are displayed, blue for FCI, red for ICI and green for MWI. The full lines correspond to the noERR experiments and are the same on the three panels. Then in addition, the mEXP curves are displayed as discontinuous lines: dashed line for mRT (left panel), dotted line for mMOD (middle panel) and dash-dotted line for mALL (right panel). The few large dots represent non-significant differences and can be found in lower layers. The left panel displays together noERR and mRT, the middle panel displays together noERR and mMOD, and the right panel displays together noERR and mALL. Colored areas highlight the differences between full and discontinuous curves. When they are not overlayed, only baseline colors appear (cyan for noERR, yellow for mRT, magenta for mMOD) and this means the perturbation has an impact on the synergy. When they are overlayed, only mixed colors appear (green, blue, dark blue) and this means the perturbation has little impact on the synergy.

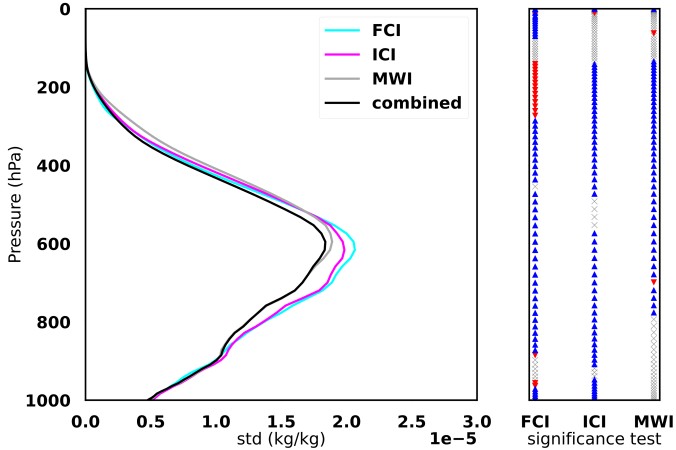

**Figure 6.** Same as figure 4 for snow retrievals



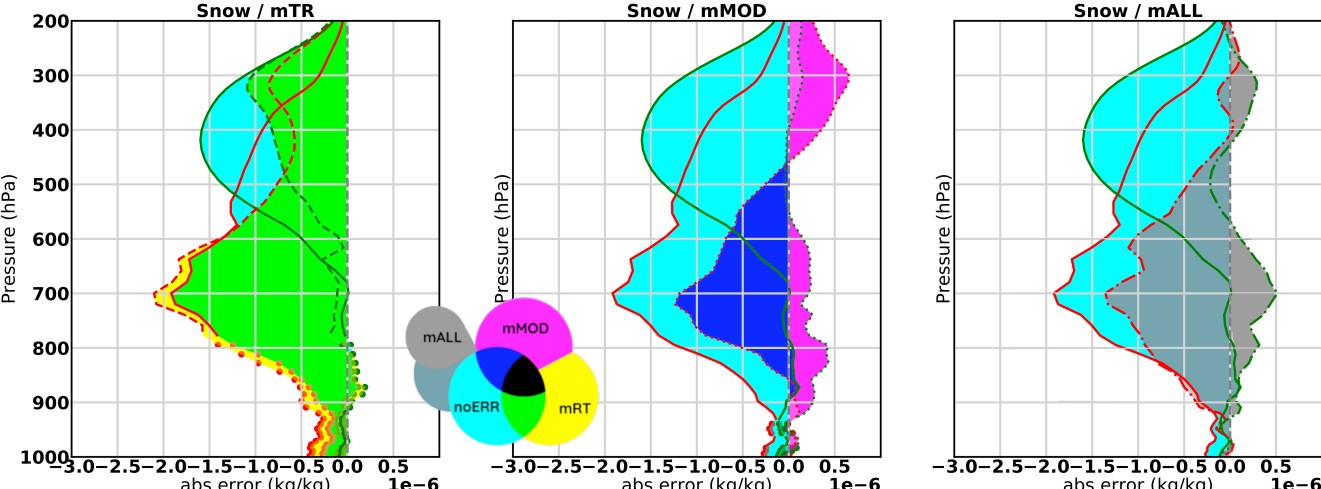

**Figure 7.** Same as figure 5 for snow retrievals

### 4.2.2 Impact of perturbations on synergy

As for snowfall, the conclusions which can be drawn are consistent with the ones for CIW: (i) the mRT perturbations have
rather small impact on the synergy, with green areas dominating the left panel, (ii) the mMOD perturbations lead to a negative
contribution to the synergistic retrievals, with blue areas not overlaying the cyan areas again, (iii) when the sources of pertur-
bations are combined, the mMOD ones remain dominant. Note that on Figure 7, the green curves for FCI are not displayed
because this instrument is not expected to well retrieve precipitation and therefore the synergy with microwave data is always
very beneficial whatever error sources are introduced.

In order to obtain such information, we consider the standard deviation, the difference of standard deviations and its area as
defined in 2.5.

### 4.3 Impact of combination and perturbations on graupel

Results for graupel retrievals are shown in Figures 8 and 9.

### 4.3.1 Impact of infrared and microwave combination

The best retrievals of graupel profiles are derived from the MWI instrument, followed by ICI. FCI retrieves graupels with an
error two times larger than the one with MW instruments which it is expected. Compared to the snow inversion (see Figure 6),
the FCI inversion is of much worse quality and this affects much more the combined inversion. A possible explanation would
be that graupels occur in convective situations with clouds even more opaque to the IR spectrum than stratiform situations.



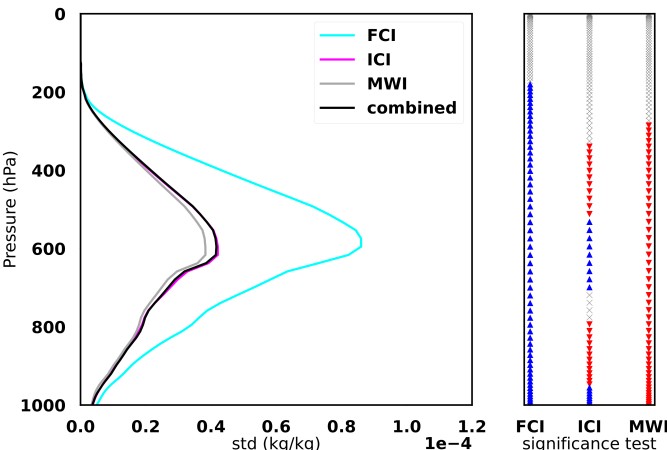

**Figure 8.** Same as figure 4 for graupel retrievals

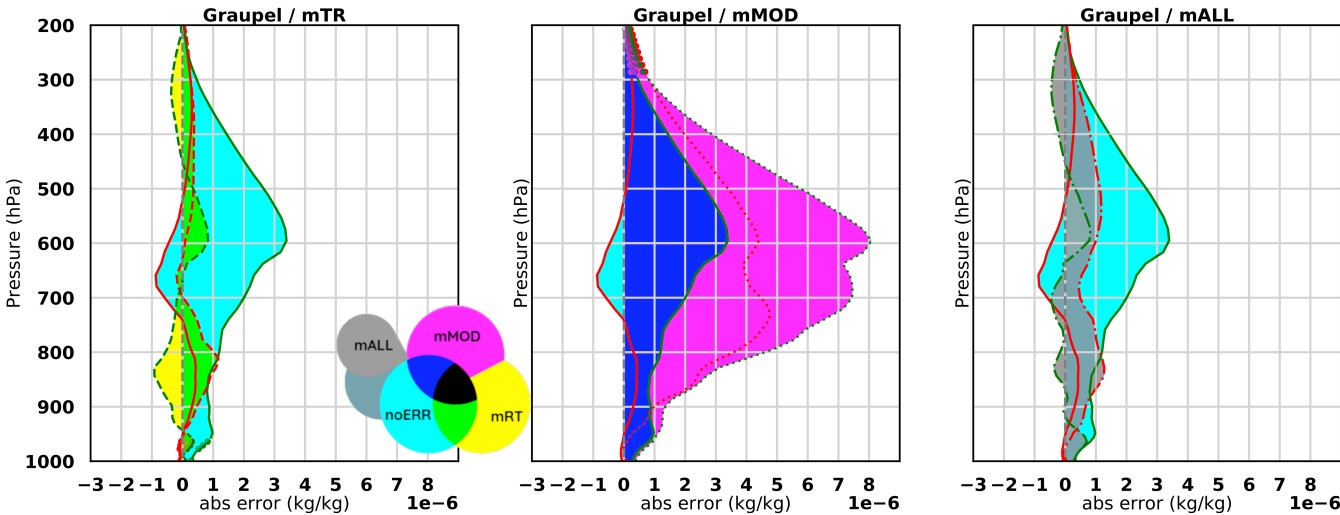

**Figure 9.** Same as figure 5 for graupel retrievals

Graupel retrievals from combined instruments are close to retrievals obtained using ICI frequencies. Overall, the combined inversion has larger errors than each MW instrument because of the negative influence of FCI. However, we have chosen to keep FCI into account in the combined inversion to remain consistent between all hydrometeors and to study the synergy between IR and MW.



### 4.3.2 Impact of perturbations on synergy

For graupel retrievals, the RT modifications appear to have a non negligible impact as only a small fraction of the figure 9 is green. Compared to the impacts obtained with mRT for snowfall retrieval, it can be explained by the choice of the particle shape used for RT perturbation. As it can be seen in Figure 9.a of Geer et al. (2021) showing the obtained BT of a simulated ice or snow cloud using different hydrometeor shapes, the perturbations introduced in particle shape (see Table 9) for graupels seem to lead to more differences for all frequencies than the ones introduced for snow.


As for CIW and snowfall, mMOD leads to a negative contribution to the synergistic retrievals for graupel with the magenta coloured areas exceeding the blue one.

When perturbations are combined in the mALL experiment, the statistics look similar to the mRT experiment, which indicate
that in this case the radiative transfer perturbations tend to dominate the impact. One interpretation of the smaller impact of the model perturbations on graupel is that the perturbations related to convective hydrometeors in our experiments are linked to the downdraft and entrainment/detrainment rate. This quantities are less directly related to the specific content of hydrometeors than the other perturbations applied to snow and CIW such as auto-conversion rates.

## 5 Discussion

To understand the impact of the synergy between IR and MW data and the uncertainties existing in NWP and RT models, we defined a step by step approach, beginning with an error free framework in order to estimate the best possible retrieval, then progressively introducing errors, in the radiative transfer both in the IR and MW simulations, and in the model as well. This process allowed to compare the impacts of those two sources of errors.

### 5.1 IR/MW synergy

In most of the experiments performed, a synergy was obtained for the three frozen hydrometeors type, thanks to the inversion algorithm which is able to find a compromise between IR and MW channels along pressure levels, using the strength of each sensor for each hydrometeor type. Even though perturbations have some impacts on retrievals, the combination of IR and MW observations remains more beneficial for the retrievals quality than using them separately in cloudy situations. Even if the FCI instrument seems to provide the best CIW retrievals and the MWI instrument the best precipitation retrievals, the new
sub-millimetric frequencies from ICI were found to well perform for both retrievals.

### 5.2 Relative importance of RT versus NWP uncertainties

Perturbations introduced in the RT model (mRT) were shown to have less impact on the retrievals than perturbations introduced in NWP model (mMOD) for CIW and snow (solid stratiform precipitation). This predominance of one type of perturbation



is independent of the spectral range. For the specific case of graupel (solid convective precipitation), the opposite result was

found. However, it is likely that the smaller effect of model perturbations for graupel retrievals is related to the perturbations of the convection scheme which do not affect directly the specific content of hydrometeors like other model perturbations do for cloud ice and snowfall.

## 5.3   Framework limitations

Realistic settings were used to introduce perturbations covering several sources of uncertainties in the inversions. The general

framework was validated by computing first guess departure statistics as function of symmetric cloud predictors both in the IR and the MW and their magnitudes were found to be compatible with the ones found in the literature with real observations. However, the applied perturbations may cover only partially the uncertainties and inconsistencies that can be encountered in the treatment of real observations:

1. Regarding perturbations of the RT model in the IR, the use of the two schemes of Baran (Baran et al., 2014) and Baum

(Baum et al., 2011) certainly do not encompass the complex variability of ice crystals in nature. A similar comment can be made regarding the perturbations in the MW simulations for which single particle shapes have been used in each simulation (Barreyat et al., 2021).

2. Regarding model perturbations, the Météo-France operational framework of perturbations, known as RPP, was used. Compared to other perturbation methods (e.g. Stochastically Perturbed Parameter Tendency (SPPT) method used at

ECMWF) to describe uncertainties in the model, the RPP method is known to lead to a rather small spread of forecasts.

3. Regarding the subgrid cloud variability representation, no modifications to the RT nor the NWP model were performed, however this source of error is of equal importance in the model and in the radiative transfer.

4. As mentioned above, the observation's geometry and resolution of each instrument was not taken into account in the framework and can be an important source of uncertainty as well, although mitigation strategies such as superobbing to

a common resolution exist to overcome part of the inconsistencies between IR and MW data.

## 5.4   Perspectives

The above mentioned limitations could be overcome by exploring larger perturbations on both RT simulations and NWP model forecasts. However, this first set of experiments indicates that the fine tuning of RT properties between IR and MW spectral ranges does not seem critical compared to the model parameterization uncertainties. It has been shown that a synergy between

the two types of datasets can still be obtained. Therefore, the next step will be to explore the use of cloudy IR data within the 4D-Var assimilation system of Météo-France which already make use of MW cloudy and precipitating data. As a first step, imagers onboard geostationary satellites will be studied and the work will then be further extended to hyperspectral instruments.



*Code and data availability.* The radiative transfer model RTTOV v13 used in this paper is available for free from https://nwp-saf.eumetsat.int/
(last access: 30 November 2022) to registered users. The numerical weather prediction model ARPÈGE is developed at Météo-France.
Datasets produced during the course of this study (ARPÈGE analyses and forecasts) are too large to be publicly archived. All model and
experiment data have been archived on the Météo-France mass storage system and can be obtained from the first author upon request.

## Appendix A: Levene's significance test

The purpose of the Levene test is to determine rather a number of sample has an equal variance. It has been published in Levene
(1960) and extended in Brown and Forsythe (1974) to use the median. It is mathematically defined as:

$$W = \frac{(N-k)}{(k-1)} \frac{\sum\limits_{i=1}^{k} N_i (\overline{Z}_{i.} - \overline{Z}_{..})^2}{\sum\limits_{i=1}^{k} \sum\limits_{j=1}^{N_i} (Z_{ij} - \overline{Z}_{i.})^2} \tag{A1}$$

where $Z_{ij} = |Y_{ij} - \tilde{Y}_i|$ with $\tilde{Y}_i$ the median of the $i$ sample, $\overline{Z}_{i.}$ are the means of the $Z_{ij}$ and $\overline{Z}_{..}$ is the overall mean of the $Z_{ij}$.

The aim of this test is to know if the variance of several samples is equal or not.
The significance level is noted $\alpha$. It is usual equal to $\alpha = 0.05$. The variances is considered non equal if:

$$W > F_{\alpha, k-1, N-k} \tag{A2}$$

where $F_{\alpha, k-1, N-k}$ is the upper critical value of the F distribution with $k-1$ and $N-k$ degrees of freedom at a significance
level of $\alpha$.

*Author contributions.* EV, PC and NF contributed to the conceptualization, formal analysis, methodology, visualization and writing - review
and editing. EV wrote the original draft.

*Competing interests.* The authors declare that they have no conflict of interest.

*Acknowledgements.* This research is funded by Météo-France and Région Occitanie (PhD grant for Ethel Villeneuve). The authors acknowl-
edge the Centre National d'Études Spatiales (CNES) for the financial support of this scientific research activity part of the Infrarouge,
Micro-Ondes et Transfert radiatif ensembliste pour la prévision des Extrêmes de Précipitations (IMOTEP) project.
Eric Defer, Laurent Labonnote and Jérôme Vidot are acknowledged for their advises on the results' interpretation. Laurent Descamps is
acknowledged for providing information and literature on RPP perturbation method.



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
