# Peer review of "Synergistic approach of frozen hydrometeor retrievals: considerations on radiative transfer and model uncertainties in a simulated framework"

_EGUsphere, 2023_

## Author Comment (AC1)

**RESPONSE TO COMMENTS**

Synergistic approach of hydrometeor retrievals: considerations on radiative transfer and model uncertainties in a simulated framework
Ethel VILLENEUVE, Philippe CHAMBON & Nadia FOURRIE

**13/04/2023**

**AUTHORS' RESPONSE**

Thank you for your review that helps us to improve the original quality of the manuscript. Your suggestions were constructive and have been incorporated into the manuscript.

**CHANGES IN THE MANUSCRIPT**

The modifications in the manuscript appear in blue in the track-change file so they can easily be monitored. Some modifications appear in red and are related to the other reviewer's comments. Some others appear in green that are minor corrections from the authors.

The reviewer's suggestions were included correcting some phrases and adding some details. Tables (3, 4 and 5 ;  6, 7 and 8) were merged to ease readability (now Table 3 and 4).

The y-axis of figures 2 and 3 was modified to fit the plots as suggested.

**COMMENTS FROM REFEREE #1**

**Review of Villeneuve et al. – Synergistic approach of hydrometeor retrievals: considerations on radiative transfer and model uncertainties in a simulated framework**

The authors present a synthetic study of retrieval synergy between microwave, infrared, and sub-mm observations for constraining ice hydrometeors. This paper is well-written and

relevant for publication in this journal. The study is well-constructed, the methodology is explained well, and reasonable conclusions are drawn about the synergistic value of these observations for frozen hydrometeors, while being clear about the caveats and shortcomings of this synthetic approach. My recommendation is for publication after some minor corrections.

**Thank you for acknowledging our work. The recommended corrections were added to the manuscript as shown below.**

One area where the manuscript is a little lacking regards the context of other synergistic studies in the literature. For instance, there are recent studies examining the combined use of MWI and ICI (https://amt.copernicus.org/articles/13/4219/2020/) and also active sensors, including using real observations (https://amt.copernicus.org/articles/15/677/2022/). As their conclusions regarding the importance of particle shape and RT errors in general are similar to the ones of this study, it is perhaps worth mentioning in the discussion section or the introduction to provide some context for readers. Other studies have also probed the importance of microphysical assumptions for different wavelengths (https://amt.copernicus.org/articles/14/5369/2021/ https://amt.copernicus.org/articles/13/501/2020/), again with conclusions that seem compatible with this study. There are also Bayesian-based studies on ice hydrometeor retrieval synergy (https://amt.copernicus.org/articles/15/927/2022/), including several others that examine radar/radiometer synergy from CloudSat and GPM. The authors' paper is more focused on eventual assimilation, but some context on retrieval focussed studies could still be helpful. The papers listed above are just from AMT, and surely there are others elsewhere.

**Thank you for providing more resources on the subject. As you suggested, some context is added to the introduction section :**

**L.35: "Several studies had already highlighted that a synergistic use of microwave or sub-millimetric radiometers and radar data was beneficial to retrieve ice hydrometeors (Pfreundschuh et al., 2020, 2022)"**

L41: "Several studies have probed the impact of different particle shapes or particle size distribution (PSD) on ice hydrometeors retrievals (e.g. Ekelund et al., 2020; Pfreundschuh et al., 2020; Geer, 2021) showing that the retrievals are sensitive to microphysical schemes"

Specific comments:

L11 – "takes advantage of both spectral range strengths" could perhaps be rewritten as "takes advantage of the strengths of both spectral ranges"

Manuscript change : from "takes advantage of both spectral range strengths" to "takes advantage of the strengths of both spectral ranges"

L13 – This last sentence is not specific enough and may need to be reworded. For instance, does "the errors" mean "the radiative transfer and numerical modelling errors"? Does "their combination" mean the combination of different sensors? It's also not clear that this is true in "all cases explored" because the graupel combined retrieval performed worse than MWI when it came to graupel.

Yes, the errors mentioned come from two sources : the radiative transfer and the numerical modeling. It has been added to the manuscript L13. The last sentence was to point out that the positive compromise created by the combination of infrared and microwave sensors remains even when considering the sources of errors. Thank you for pointing it out. I clarified it in the manuscript.

L21 – "a significant information content" is quite vague, suggest rewording

Change : this wide range of frequencies is characterised by a large diversity of information a significant information content on all hydrometeor phases along the vertical

L26 – Worth spelling out what all-sky means for readers

Added : All-sky observations, in contrast to clear-sky observations, gather all meteorological situations, whether it is cloudy or not.

L33 – Suggest changing "at discussing" to "to explore"

Changed "at discussing" by "to explore".

L50 – This is nitpicking, but sub-mm is greater than 300GHz, so ICI will technically measure MW and sub-mm wavelengths and L51 – Again a technicality, but there have been

short-lived sensors measuring at sub-mm, and AWS might be launched before ICI, so this statement could be toned down.

Change about ICI : "the EPS-SG-Ice Cloud Imager (ICI) with sub-millimetric frequencies (> 300  GHz)  in addition to microwave frequencies (> 183 GHz) which gives new information on ice cloud"

Table 2 – It's a bit confusing how "OBS FG" is shown. Does this mean OBS in rows and FG in columns? Could make this clearer.

**I added arrows to point where OBS and FG are.**

L113 – See tables 4 & 5, presumably?

Tables 3, 4, 5 – Would it be possible to combine into one table? This would make it easier to compare values across sensors.

**Tables 3, 4 and 5 are now merged**

L145 – Tables 6, 7, & 8, presumably?

Tables 6, 7, 8 – Same as above, could these be combined?

**Same, tables 6, 7 and 8 are now merged.**

L211 – It would be helpful to provide more detail here to explain exactly why this validation comparison is done. Right now it feels quite implicit and the reasoning is split up (L234, L355), but it would be helpful to spell out exactly why the validation was performed at the beginning of this section.

**I added in the text L219: "a validation metric is needed to verify the accuracy of the chosen settings of the simulations" to justify the validation at the beginning of the section**

Figures 2 & 3 – y-axis should be STD and mean?

**Yes, I corrected the y-axis.**

Figures 4, 6, 8 – Here the y-axis could be reasonably cut off at 100hPa, as presumably the significance differences are spurious noise above this level.

**Figures 4, 6 and 8 were cut off at 100hPa as suggested.**

L271 – Reword "instrumental synergy" to something like "synergy of the instruments"

Changed from "instrumental synergy" to "synergy of the instruments"

Figures 5, 7 9 – Here the x-axis label of "abs error" was quite confusing for me. Isn't this the difference definition given in Section 2.5? Also a typo in first panel of 'mTR' rather than 'mRT'

The x-axis represents "DIFF" as defined in eq. (4), the label was adjusted. Also "mTR" has been changed to "mRT".

L306 – Why is this given halfway through the results section? It would make much more sense at the beginning of Section 4.

Yes it makes more sense at the beginning of the section, this has been moved.

L318 – Worth mentioning that the FCI curves are again absent in Fig. 9, as stated for Fig. 7

Added L.347: "As for Snow retrievals, the curves for FCI are not displayed because this instrument is not expected to well retrieve this variable"

---

## Author Comment (AC2)

**RESPONSE TO COMMENTS**

Synergistic approach of hydrometeor retrievals: considerations on radiative transfer and model uncertainties in a simulated framework
Ethel VILLENEUVE, Philippe CHAMBON & Nadia FOURRIE

**17/10/2023**

**AUTHORS' RESPONSE**

Thank you for your review that highlighted points to improve the clarity of the manuscript. Your suggestions were appreciated and added to the manuscript.

**CHANGES IN THE MANUSCRIPT**

The modifications in the manuscript appear in red in the track-change file so they can easily be monitored. Some modifications appear in blue and are related to the other reviewer's comments. Some others appear in green that are minor corrections from the authors.

The reviewer's suggestions were included, the detailed response for specific comments follows.

**COMMENTS FROM REFEREE #2**

**Review comments for "Synergistic approach of hydrometeor retrievals: considerations on radiative transfer and model uncertainties in a simulated framework" by Villeneuve et al.**

This work thoroughly assessed the benefits of combining passive thermal infrared (TIR), sub-millimeter (sub-mm) and microwave (MW) instruments in retrieving frozen hydrometeors (cloud ice, snow, and graupel) using a Bayesian retrieval framework and a regional model outputs as the input "truth" for retrieval and validation. What's more, this work also evaluated two retrieval error sources from radiative transfer model microphysics assumption (mRT experiments) and from model microphysics parameterization schemes

(mMOD experiments) that might lead to degradation of the synergy. Three upcoming spaceborne instruments (FCI, ICI and MWI) are brought in for specific channel frequency settings, but their mis-match footprint, view-angle, etc., were not considered yet in this paper. The major conclusion of this work is that synergy overall is better than using any of the three instruments individually for retrieving cloud ice and snow profiles, but not necessarily for graupel. Retrieval performance (i.e., error) tends to be more sensitive to model microphysics scheme parameters than to which type of ice particle shape that is chosen (again, opposite for graupel). The discussions speculate some possible explanations for the behaviors error metrics (mainly standard deviation of inversion error compared with using single instrument). Some other error sources that are important but not considered in this work are laid out in the end.

Overall I think this is a solid paper that involves extensive efforts and a sound framework for testing and validation. Therefore, I strongly support the publication of this work eventually. However, I do think there are some important issues that need to be clarified, and some more potential experiments need to be considered. I wouldn't put a "major revision" as my recommendation, as the idea, methodology, execution and result presentation do not have serious issues and worth some great appreciation.

**Thank you for the acknowledgment of the work and your support on the publication.**

The overarching goals are a bit ambitious in design: trying to assess two important problems (i.e., synergy, and synergy sensitivity) in one paper. I would rather consider separating into two companion papers so each one can focus on one problem which gets a chance to be more elaborated. Right now the first 14 pages are spent for describing experiment settings, while Page 15 – 20 are for presenting the results, which followed with one page very brief discussion. I feel it's not an ideal paper structure. Below are my major concerns:

(1) For assessing the synergy benefit, CA and SCP are introduced in Equation (5) and (8) for IR and MW, respectively. Maybe these are some standard parameters that data assimilation people used a lot already, but for a retrieval person like me, I lost clue on why

using these two metrics, what are their physical meanings, and why it's inconsistent between TIR and MW. Substantial explanations and discussions are needed here.

FG and OBS have several sources of error. Here, we want to evaluate if the chosen assumptions are accurate or not. To do so, we use data assimilation metrics. The assumptions we want to evaluate are the simulation of hydrometeors, the representation of clouds in the forecast model (using microphysical and convection parametrisations) and also the way we simulated the FG and the fake OBS, with lagged forecasts. The paragraph (i) of the section 3 was further elaborated to clarify :

L.225: "As the study is based on simulations both for observations and first guess, a validation metric is needed to verify the accuracy of the chosen settings of the simulations. Data assimilation metrics are used to validate the framework. Both FG and OBS have sources of errors, in the simulation of hydrometeors, in the representation of clouds in the forecast model (microphysical and convection parameterizations) and also the chosen assumption for the simulated OBS and FG that is a lagged forecast."

For example, in Fig. 2 why the average of errors are small positive for noERR and mRT, larger and negative for mMOD, but the standard deviations are comparable in size?

One contribution to the standard deviation is the mislocation of the clouds in the first guess compared to observations. This contribution is present in both mRT and mMOD experiments. Regarding the bias, the impact of modifications in mMOD is indeed more important on the bias than mRT, which is also consistent with the findings of the study highlighting the importance of model errors. L.258 "The modifications introduced in the model appear to have more impact on the bias than on the STD.  In the following sections, STD will be studied and the relative impact of mRT and mMOD experiments will be highlighted."

Shouldn't the CA error increases for thicker clouds (I guessed from the grey bar, which I assume corresponds to the number of cases)?

The standard deviation increases with CA until CA = 25K. This increase exists when the number of cases involved is high enough. After that, more fluctuations are noted in the std. In figure 5 of Okamoto, 2017, we can see the same trend for the band 13 (10.4µm) of AHI.

[Figure]

**Figure 5.** O−B SD and mean (or bias) as a function of CA at bands (a) 8, (b) 10, (c) 13, and (d) 16 in CASE1. O−B SD calculated from samples passing the homogeneity QC and all the three QCs are plotted with thin and thick red lines, respectively, for the left axis. O−B biases are plotted with blue lines for the left axis. Light (dark) grey bars are the log number of samples used after the homogeneity QC (all the three QCs) on the logarithmic scale at the right axis. Black lines represent a linear function (Eq. (4)) for predicting O−B SD with CA estimated from samples after the three QCs in CASE1, CASE2 and CASE3.

L.258 "For CA > 30 K, the number of cases decreases and more fluctuation on STD and bias appear. Okamoto et al. 2017 highlighted that this decreasing is due to the number of cases that is too small to be significant."

Using correction and quality control for data assimilation, this can be flattened as shown on figure 9 of Okamoto et al., 2021.

[Figure]

**FIGURE 9** (a–d) The SD (red) and mean (blue) O − B as a function of $C_A$ for BT at bands 8, 9, 10 and 13 of Himawari-8/AHI. The statistics were calculated from the samples in all-sky conditions that passed the QC from 1 to 31 August 2018. The number of samples is plotted as grey bars on the right axis on the log scale. The black lines indicate the observation error models representing the O − B SD as a linear function of $C_A$ [Colour figure can be viewed at wileyonlinelibrary.com]

We did not apply any quality control in our experiments but more work about infrared data assimilation should focus on that.

L.263 "A quality control is added in the study of Okamoto et al. (2021) (Figure 9 (d)) that flatten the magnitude of STD. Further exploration on a quality control for data assimilation in ARPEGE model could be done in a future study to investigate these results."

Would you concern about using STD difference to assess your synergy performance when the bias are not even the same sign?

The bias was indeed not the main focus of our study because the magnitude is  much lower than the std. Moreover, as this study aims to be used for data assimilation, bias could be corrected with bias correction in the DA system. This was mentioned in the text

L.203 "The bias will not be shown as it is overall smaller than the std values in most of the experiments."

and added L.204 : "Moreover, in a data assimilation system, a potential bias could be corrected a posteriori."

(2) For mMOD experiments, I never understand how tuning so many parameters can give you one final assessment value at the end? Did you carry out two sets of experiments, one using the lower values in your Table 11, one using the higher bound values, and then computing their difference against your noERR results?

As acknowledged in this paper, it is expected to see significant compensations among different parameters. It worths at least one paragraph here or better an appendix section to describe details about mMOD experiment settings.

mMOD perturbations are made with a random draw from uniform distribution. Any value in the minimum-maximum ranges defined by tables 7 and 8 can be randomly chosen. 1 value is chosen per day. The extreme values are not especially tested.  To make it clearer for the reader, one sentence was added in section 2.4.2. L189 : Any value between the minimum ($X_{MIN}$) and the maximum ($X_{MAX}$) values could be chosen to replace the default (noERR) value.

(3) I'm having some issue with the settings of mRT experiments, in particular, the selection of snow particle shape for noERR and mRT. 183 GHz and sub-mm channels are .

particularly sensitive to snow particle shape, and many previous observations using limited field campaigns or satellite observations have demonstrated that it's more proper to use "Evans snow" or "Liu's sector snow" for snow, and the largest discrepancies come from "soft spheroid" (e.g., Ekelund et al., 2020; Gong et al., 2021). The two snow shapes usually produce quite similar results for sub-mm and MW channels. This is the part that I feel uncomfortable for the settings and suggest to change. For "graupel", usually we use "8-column aggregates" in ARTS, but I guess that's not a serious issue as I expect sub-mm and high-frequency MW bands to be saturated to graupels quickly anyway.

Other choices could have been possible. We chose settings used for operational configuration. Different options could be tested in the future including your suggestion for high microwave frequencies. One sentence was added in the section 2.4.1 ("L 179 Other choices would have been possible using recent studies such as Ekelund et al. (2020) or Gong et al. (2021), that suggest other particles for frozen hydrometeors for MW and sub-mm frequencies.") and the perspectives of the paper to explain that the methodology applied is general and that other RT choices could have been made.

I'm not surprised to see graupel retrieval error is not sensitive to mMOD, and agree with the authors that cumulus parameterization scheme is probably that matters for graupel instead of microphysics details in cloud ice and snow (still surprised me that tuning entrainment rate seems to not work). However, it's also worth noting in the paper that none of these channels are really sensitive to graupel, so it's expected that the synergy of the three would get things worse. I might have overlooked this point in the manuscript, or I feel it is not emphasized enough in the discussion related to Fig. 8 & Fig. 9.

The instrument that would retrieve the best graupel is MWI. The other instruments add less relevant information as the higher frequencies are sensitive to smaller crystals. That is what we can see on figure 8.

(4) Another relatively important issue that was overlooked in mRT experiment is the assumption of particle size distribution (PSD), which matters a lot for sub-mm and MW channels. There is a variety of PSD choices in ARTS, and I think it worths to be considered in the mRT experiments.

mRT experiment is based on modifications on the particle shape mainly. However, for the IR, Baum and Baran schemes involve indirect change in PSD via the modification of the mass-dimension relation. For MW, PSD parameters are modified for ice crystal. We use a modified gamma distribution for both, with different values for mu and Lambda. For Snow and Graupel, PSD remains the same (Field et al., 2007) but the change in the shape of the particle involves change in the diameter that affects the PSD. To make it clearer for the reader that not only the crystal shape has been changed, one sentence was added for both IR and MW in section 2.4.1.

L.173 "The Table 5 shows the modification in terms of particle shape. The PSD is also modified between these two versions, using different values for the parameters of the modified gamma distribution."

L.176 "Here, PSD are indirectly modified as the change between Baran and Baum scheme involve modifications of the mass-dimension relation of hydrometeors."

Minor issues:

Fig. 5, 7 & 9: I'd strongly suggest you to include a mean IWC profile with standard deviation as a reference panel in addition to the current ones. This is helpful for readers to at least visually assess the percentage error of improvements.

Figures were added as figures 4, 7 and 10 to show observation mean profiles for each hydrometeor (black line) with +/- STD (grey area).

CIW :

[Figure]

Graupel :

[Figure]

Snow :

[Figure]

For example, I found it's very interesting to see mRT improves cloud ice retrieval when you focus on snow (Fig. 7a, cyan vs. green). By the way, mRT seems to be mis-spelled as "mTR" in all three figure sub-titles.

Above 600 hPa we notice that mRT modifications improve snow retrievals. However, at this altitude, the snow content has low values.

One sentence was added to the manuscript to stress out that this aspect requires further investigation.

L.324 "On mRT panel, we can notice that mRT modifications seem to improve snow retrievals above 500~hPa. Further exploration could allow to elucidate that comment, by testing more particle shapes or identifying in which situations this improvement occurs."

Section 5.3: Can't agree you more with your point #3 and #4. For #3, please consider citing Barlakas and Eriksson (2020). It's a nice paper focusing on sub-grid variability for sub-mm radiometer retrievals. For models with 5 km resolution, it's comparable to footprint size of these sensors but facing similar order of sub-grid variability. For #4, it is real when it comes to the real algorithm design for combined algorithms, which worths another paper to discuss and #3 and #4 are tightly tied. (guess this is just my comment)

Indeed, conclusions 3 and 4 are linked. As you suggested, Barlakas and Eriksson 2020 are cited to add a link between the two points.

L.395 "Barlakas and Eriksson (2020) focused on sub-grid variability of sub-mm frequencies and highlighted that the instrument's footprint has impact on the model's uncertainties. R2As mentioned above, the observation's geometry and resolution of each instrument was not taken into account in the framework. For future studies, the instruments' footprints could be taken into account to investigate the model error induced by the sub-grid cloud representation."

References:

Barlakas and Eriksson (2020): **https://doi.org/10.3390/rs12030531**

Ekelund et al. (2020): https://doi.org/10.5194/amt-13-501-2020

Gong et al. (2021): https://doi.org/10.5194/essd-13-5369-2021